# Invariance-Based Causal Estimation in the Presence of Concept Drift

**Katie Matton**[1, 2]                **John Guttag**[1]                **Rosalind Picard**[2]

[1]CSAIL, Massachusetts Institute of Technology, Cambridge, Massachusetts, USA
[2]MIT Media Lab, Massachusetts Institute of Technology, Cambridge, Massachusetts, USA

## Abstract

Machine learning models are prone to relying on spurious correlations. Recently, there has been substantial progress towards solving this problem using invariant learning methods. These methods exploit the invariance of causal mechanisms across environments to distinguish between causal and spurious parts of the feature space. Existing methods have produced impressive results in constrained settings, but rely on assumptions that limit their applicability to real-world problems. In this work, we relax one of these assumptions: the absence of concept drift. We examine a simple case of concept drift, in which the label distribution is influenced by environment-dependent additive shifts. We show that in this setting, existing methods fail. We then present a new method, called alternating invariant risk minimization (AIRM), that solves the problem. It works by alternating between using invariant risk minimization to learn a causal representation, and using empirical risk minimization to learn environment-specific shift parameters. We evaluate AIRM on two synthetic datasets, and show that it outperforms baselines.

## 1 INTRODUCTION

A problem with most machine learning models is that they incorporate *associations* rather than *causal relationships*. This makes them prone to relying on spurious correlations. Spurious correlations offer a tempting "short-cut" to inflate training performance, but do not reliably hold in new settings. Hence, models that exploit them tend to have poor generalization performance.

This problem can have alarming real-world consequences. A recent study found that models trained to detect COVID-19 from chest X-rays learned to rely on correlated artifacts of the data collection process rather than anatomical information DeGrave et al. [2021]. Despite their strong training accuracy, these models failed when tested in new hospitals. Similar results have also been found for skin cancer screening Winkler et al. [2019] and pneumonia detection Zech et al. [2018].

To solve this problem, machine learning researchers have begun to turn to ideas from causal inference. One idea exploits the connection between invariance and causality Neuberg [2003], Bühlmann [2020]. Whereas spurious correlations are often unstable across settings, causal relationships remain invariant across different experimental settings. Exploiting this has led to a new set of causal estimation methods that work by identifying invariances across data collected from heterogeneous environments. This line of work was pioneered by Peters et al. [2016], which introduced Invariant Causal Prediction (ICP), a method that works by using statistical hypothesis tests to identify invariances. Other studies followed, including Invariant Risk Minimization (IRM) Arjovsky et al. [2019], which enforces invariance using an optimization-based approach. IRM was one of the first to use the principle of invariance to learn causal representations within deep networks, and it has inspired many further studies that analyze or extend it Ahuja et al. [2020], Rosenfeld et al. [2020], Yin et al. [2021], Ahuja et al. [2021], Liu et al. [2021]. Collectively, these works show that invariant learning methods are useful for reducing the reliance of machine learning models on spurious information.

Although there has been substantial progress in the area of invariant learning, existing methods make restrictive assumptions that often do not hold in the real-world. In this work, we focus on alleviating one of these assumptions: the absence of concept drift across data collection environments. Concept drift refers to a change in $\mathbb{P}(y|x)$, the relationship between the input and target variables Tsymbal [2004]. It is often induced by a *hidden context* variable. For example, the outcome for a hospitalized patient might depend upon both documented patient data and unmeasured factors, such as the quality of the caregivers. When collecting data

*Accepted for the 38th Conference on Uncertainty in Artificial Intelligence* (UAI 2022).

from multiple environments, it is quite plausible that they have different hidden contexts, and therefore exhibit concept drift. Notably, this drift can occur even when we condition solely on the subset of $x$ that are *causes* of $y$. Such drift directly violates the main invariance assumption underlying existing methods. This leads us to the question: *Can invariance-based methods still be used to estimate causes in the presence of concept drift? And if so, how?*

In this study, we make progress towards answering this question. We consider a simple example of concept drift, in which a single environment-dependent hidden variable influences the label distribution through a constant-mean shift. We show that even in this simple setting, existing invariant learning methods fail. To solve this problem, we introduce a new method, called alternating invariant risk minimization (AIRM). AIRM works by alternating between two optimization phases: it uses IRM to learn a causal representation, and then it uses empirical risk minimization (ERM) to learn the environment-specific shift parameters. Through synthetic data experiments, we show that AIRM is able to distinguish between spurious and causal variables, and outperforms baselines on out-of-distribution (OOD) adaptation problems.

## 2 PROBLEM SETUP

To make the problem setup clear, we begin with a motivating example. Consider the task of predicting skin cancer risk from dermoscopic images. Assume that we collect data from multiple dermatology practices, and that our goal is to use this data to train a model that works well on data collected from future, unseen dermatology practices. In order to achieve this, we desire a model that avoids relying on spurious correlations. For example, in this domain, there is a known problem with models learning to use the presence of inked markings as a feature Winkler et al. [2019]. Dermatologists are typically more likely to add these markings if they believe the lesion may be cancerous, but since markings are not causally related to skin cancer, this correlation cannot be depended upon to hold in all settings.

For ease of understanding, first consider a simplified version of this problem. For each dermoscopic image, we observe three variables: $x_1$, a continuous variable representing the abnormality of the lesion; $x_2$, a binary variable indicating the presence of markings; and $y$, a continuous cancer risk score assessed by a dermatologist. For now, we let $\boldsymbol{x} = < x_1, x_2 >$. Later, we examine the more difficult setting in which $\boldsymbol{x}$ is an image that indirectly contains $x_1$ and $x_2$.

We consider datasets of the form $D = \{(\boldsymbol{x}, y)\}$ that come from different dermatology practices, or training environments $e \in \mathcal{E}_{\text{tr}}$. We aim to learn a predictive function $f : \boldsymbol{x} \to y$, which performs well on unseen test environments $e \in \mathcal{E}_{\text{test}}$. We examine two variations of this problem.

In the first, we make assumptions that are consistent with those adopted in prior work. In the second, we relax these assumptions and introduce the presence of concept drift.

### 2.1 TRADITIONAL SETTING

We start by considering a setting that is consistent with the assumptions made in prior work.

**Example 1.** *Assume that the data is generated according to the Structural Causal Model (SCM)* Wright [1921]:

$$x_1 \leftarrow \mathcal{N}(0, \sigma_{e,x_1}^2)$$
$$y \leftarrow x_1 + \mathcal{N}(0, \sigma_{e,y}^2)$$
$$x_2 \leftarrow \text{Bernoulli}(\sigma(\kappa_e y + \mathcal{N}(0, 1)))$$

*where $\sigma_{e,x_1}$, $\sigma_{e,y}$, and $\kappa_e$ are environment-specific parameters.*

We include a visualization of the corresponding directed acyclic graph (DAG) in Figure 1a. The SCM implies that lesion abnormality, $x_1$, is a direct cause of the cancer risk assessment, $y$, and that the presence of markings, $x_2$, is a spurious variable. The SCM also implies that the relationship between $x_2$ and $y$ differs across environments, since it is modulated by the environment-dependent parameter $\kappa_e$. This encodes the assumption that in some environments, a high cancer risk assessment may increase the likelihood of markings, whereas in others it may have a negligible or inverse effect. An ideal predictive function $f$ would ignore $x_2$ and use only $x_1$.

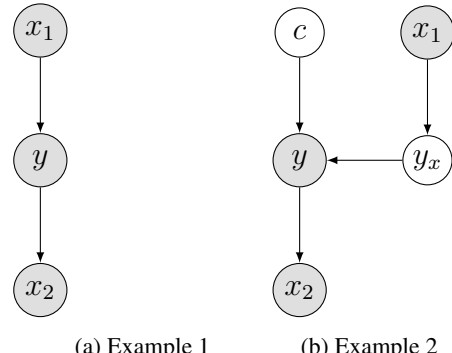

(a) Example 1          (b) Example 2

Figure 1: DAGS for two example problems.

Existing invariant learning methods, such as IRM, are able to find this solution, as we show in Section 4. While the implementation details of existing methods differ, the intuition behind why they work is largely the same. Most existing methods exploit the following invariance assumption:

**Assumption 1.** *Let $\phi(\boldsymbol{x})$ be a representation of $\boldsymbol{x}$ in which only the causal information in $\boldsymbol{x}$ is retained. Then,*

$$\mathbb{E}_e[y|\phi(\boldsymbol{x})] = \mathbb{E}_{e'}[y|\phi(\boldsymbol{x})]$$

*for all $e, e' \in \mathcal{E}_{tr} \cup \mathcal{E}_{test}$.*

This assumption is useful because it implies that one can identify the causal parts of $\boldsymbol{x}$ by searching for a representation that satisfies this invariance property. Indeed, this strategy works for Example 1. $\phi(\boldsymbol{x}) = x_1$, since $x_1$ is a direct cause of $y$ and $x_2$ is not. $\mathbb{E}[y|x_1] = x_1$ is the same for all environments, whereas $\mathbb{E}[y|x_2]$ and $\mathbb{E}[y|x_1, x_2]$ are environment dependent.

Example 1 serves as an illustration of how, under the right conditions, existing invariant learning methods are a powerful tool for causal estimation. We next explore what happens when the concept-drift assumption is not satisfied.

## 2.2 CONCEPT DRIFT SETTING

Consider the assumptions encoded by the DAG associated with Example 1 (Figure 1a). The assumption that all of the variables are observed simplifies the problem because it implies that $x$ contains *all* the information necessary to determine $y$, up to noise. However, there are many cases where this is not true. For example, in the skin cancer prediction problem, there are multiple factors that may influence a dermatologist's assessment of cancer risk that are *not* observable in the dermoscopic images. These include the clinical training received by the dermatologist, the dermatologist's level of risk aversion, and the dermatologist's knowledge about the patient's medical history.

To encode this understanding, we introduce a new variable $c$, which represents the hidden context that influences a dermatologist's assessment of cancer risk given dermoscopic images. The distribution of $c$ is likely to vary across environments, e.g., dermatology practices. Since $c$ causally influences $y$, when the distribution of $c$ is different across environments, the conditional probability of $y$ given its *observed* direct causes ($x_1$), will also differ. In other words, $c$ induces a *concept drift* across environments. The presence of this type of concept drift directly violates Assumption 1. Since most existing invariant learning methods rely on Assumption 1, they do not work in this setting, as we now show.

**Example 2.** Assume the following SCM:

$$x_1 \leftarrow \mathcal{N}(0, \sigma_{e,x_1}^2)$$
$$y_x \leftarrow x_1 + \mathcal{N}(0, \sigma_{e,y}^2)$$
$$c \leftarrow \mathcal{N}(\mu_e, \sigma_{e,c}^2)$$
$$y \leftarrow y_x + c$$
$$x_2 \leftarrow \text{Bernoulli}(\sigma(\kappa_e y + \mathcal{N}(0, 1)))$$

*where $\sigma_{e,x_1}$, $\sigma_{e,y}$, $\sigma_{e,c}$, $\mu_e$, and $\kappa_e$ are environment specific parameters.*

The DAG corresponding to Example 2 is shown in Figure 1b.

This problem is similar to Example 1, with several differences. We introduce the variable $y_x$ to represent the best assessment of cancer risk given *just the lesion abnormality score*. Whereas in Example 1 this quantity was the observed variable $y$, we now assume that it is a latent variable that influences the observed labels. We use the variable $c$ to represent the hidden contextual factors that, in addition to $x_1$, influence the dermatologist's assessment of cancer risk $y$. We assume that the observed label $y$ is the sum of the cancer risk associated with the lesion abonormality score $y_x$ and the risk attributed to contextual factors $c$.

In this example, $\phi(\boldsymbol{x}) = x_1$ since it is still the case that $x_1$ is the only observed cause of $y$. However, the conditional probability of $y$ given $\phi(\boldsymbol{x})$ now varies across environments; i.e., $\mathbb{E}[y|\phi(\boldsymbol{x})] = \mathbb{E}[y|x_1] = x_1 + \mu_e$. Hence, Assumption 1 no longer holds. As we verify empirically in Section 4, existing invariant learning methods that rely on Assumption 1 fail to solve this problem. We describe a new method for solving this problem in the next section.

## 3 OUR METHOD: ALTERNATING IRM

We introduce a new method for causal estimation from multi-environment data that works even in the presence of certain types of concept drift. We constrain the set of concept drifts with the following assumptions:

**Assumption 2a.** *Let $\phi(\boldsymbol{x})$ be a representation of $\boldsymbol{x}$ in which only the causal information in $\boldsymbol{x}$ is retained. Then there exists a latent variable $y_x$ such that:*

$$\mathbb{E}_e[y_x|\phi(\boldsymbol{x})] = \mathbb{E}_{e'}[y_x|\phi(\boldsymbol{x})]$$

*for all $e, e' \in \mathcal{E}_{tr} \cup \mathcal{E}_{test}$.*

Assumption 2a represents the idea that if we were able to remove the effect of the latent context variable, we would still expect the relationship between $y$ and the causal covariates to be invariant.

**Assumption 2b.** *Let $c$ be a random variable that is distributed according to an environment-specific normal distribution: $c \sim \mathcal{N}(\mu_{e,c}, \sigma_{e,c}^2)$. Then,*

$$y = y_x + c$$

With Assumption 2b, we limit the set of latent context variables we consider to those that act by shifting the conditional label distribution up or down. While this is reasonable in some scenarios (e.g., risk labels produced by dermatologists with different levels of risk aversion), it may be too simplistic in others. We leave the consideration of more complex contextual variables for future work.

This set of assumptions is strictly weaker than the set considered in prior work. If we assume that $c = 0$ for all environments, we recover Assumption 1.

Under these assumptions, if we knew the environment-specific means $\mu_e$ of the context variables, we could compute $y_x$ from $y$, up to noise. Then, with known $y_x$, the task of training an invariant prediction model that maps from $\boldsymbol{x}$ to $y_x$ is exactly the problem considered in the traditional setting (see Section 2.1), which can be solved with existing methods. Conversely, if we knew the correct causal model mapping $\boldsymbol{x}$ to $y_x$, we could estimate $\mu_e$ from $y$ and $y_x$, up to noise. Our method, called Alternating Invariant Risk Minimization (AIRM), is based on this intuition.

We include pseudocode describing our method in Algorithm 1. It takes training data from a set of environments $e \in \mathcal{E}_{\text{tr}}$ and the loss function $\ell$ to minimize as input. It produces two sets of parameters: (1) $\boldsymbol{\phi}$, the parameters of a function $f(\boldsymbol{x}; \boldsymbol{\phi})$ that maps from $\boldsymbol{x}$ to predictions of the latent variable $y_x$, and (2) $\{\beta_e\}_{e \in \mathcal{E}_{\text{tr}}}$, intercept terms used to capture the effect of the hidden context variables. The resulting predictions for environment $e$ are generated as $\hat{y} = f(\boldsymbol{x}; \boldsymbol{\phi}) + \beta_e$.

AIRM estimates parameters by alternating between two phases of optimization:

1. Given the current estimate of each $\beta_e$, it applies IRM to estimate $\boldsymbol{\phi}$, considering $y - \beta_e$ as the targets.

2. Given the current estimate of $\boldsymbol{\phi}$, it applies empirical risk minimization (ERM) to estimate the $\beta_e$ parameters. This can be done using SGD. Alternatively, when using mean squared error (MSE) loss, we can compute these estimates directly as $\beta_e = \frac{1}{n_e} \sum_{i=1}^{n_e} y_i - f(\boldsymbol{x}_i; \boldsymbol{\phi})$.

There are four hyperparameters: $\lambda$ the weight of the invariance penalty of IRM, $k_1$ the number of SGD steps to take for each IRM optimization phase, $k_2$ the number of SGD steps to take in each ERM optimization phase, and $\eta$ the learning rate.

Our method is designed for the domain adaptation evaluation scenario. While the invariant prediction model $\boldsymbol{\phi}$ can be directly applied to new environments, to estimate $\beta_e$ we need environment-specific data. Therefore, we assume that for each test environment $e \in \mathcal{E}_{\text{test}}$, we have a small amount of adaptation data that can be used to learn $\beta_e$. Given a trained model $\boldsymbol{\phi}$ and data from a new environment $e$, we estimate $\beta_e$ using ERM.

## 4 EXPERIMENTS

We evaluate AIRM with two experiments conducted on synthetic data. The first experiment directly follows from the example problems described in Section 2. The second uses a slightly more challenging example in which the inputs are images.

We compare AIRM to the baselines listed below. Each

---

**Algorithm 1** AIRM

**Input:** $D^e = \{(\boldsymbol{x}, y)\}_{i=1}^{n_e}$ from $e \in \mathcal{E}_{\text{tr}}, \ell, \lambda, k_1, k_2, \eta$
**Output:** $f(\cdot; \boldsymbol{\phi}), \{\beta_e\}_{e \in \mathcal{E}_{\text{tr}}}$
Initialize $\boldsymbol{\phi}, \{\beta_e\}_{e \in \mathcal{E}_{\text{tr}}}$ randomly
  **while** not converged **do**
    fix $\{\beta_e\}_{e \in \mathcal{E}_{\text{tr}}}$     $\triangleright$ (1) IRM optimization phase
    $y_{x,i} \leftarrow y_i - \beta_e$ for $y_i \in D^e, e \in \mathcal{E}_{\text{tr}}$
    **for** step $\leftarrow 1$ to $k_1$ **do**
      $R^e(f(\cdot; \boldsymbol{\phi})) \leftarrow \frac{1}{n_e} \sum_{i=1}^{n_e} \ell(f(\boldsymbol{x}_i; \boldsymbol{\phi}), y_{x,i})$ for $e \in \mathcal{E}_{\text{tr}}$
      $\mathcal{L}_{IRM}(f(\cdot; \boldsymbol{\phi})) \leftarrow$
        $\sum_{e \in \mathcal{E}_{\text{tr}}} R^e(f(\cdot; \boldsymbol{\phi})) + \lambda || \nabla_{w|w=1.0} R^e(w \cdot f(\cdot; \boldsymbol{\phi})) ||^2$
      $\boldsymbol{\phi} \leftarrow \boldsymbol{\phi} - \eta \frac{\partial}{\partial \boldsymbol{\phi}} \mathcal{L}_{IRM}(f(\cdot; \boldsymbol{\phi}))$
    **end for**
    fix $\boldsymbol{\phi}$      $\triangleright$ (2) ERM optimization phase
    $c_i \leftarrow y_i - f(\boldsymbol{x}_i; \boldsymbol{\phi})$ for $\boldsymbol{x}_i \in D^e, e \in \mathcal{E}_{\text{tr}}$
    **for** step $\leftarrow 1$ to $k_2$ **do**
      **for** $e$ in $\mathcal{E}_{\text{tr}}$ **do**
        $R(\beta_e) \leftarrow \frac{1}{n_e} \sum_{i=1}^{n_e} \ell(\beta_e, c_i)$ for $e \in \mathcal{E}_{\text{tr}}$
        $\beta_e \leftarrow \beta_e - \eta \frac{\partial}{\partial \beta_e} R(\beta_e)$
      **end for**
    **end for**
  **end while**

---

are used to estimate a prediction function $f$. We consider two forms of $f$: (1) a global model $f_g = f_g(\boldsymbol{x}; \boldsymbol{\phi})$ and (2) a model with an environment-specific intercept $f_e = f_g(\boldsymbol{x}; \boldsymbol{\phi}) + \beta_e$.

- **ERM** minimizes the objective:

$$\sum_{e \in \mathcal{E}_{\text{tr}}} R^e(f) = \sum_{e \in \mathcal{E}_{\text{tr}}} \frac{1}{n_e} \sum_{i=1}^{n_e} \ell(f(\boldsymbol{x}), y_i) \qquad (1)$$

We let $f = f_g$ for problems without concept drift and $f = f_e$ for problems with concept drift.

- **IRM** (Arjovsky et al. [2019]) minimizes the objective:

$$\sum_{e \in \mathcal{E}_{\text{tr}}} R^e(f_g) + \lambda || \nabla_{w|w=1.0} R^e(w \cdot f_g) ||^2 \qquad (2)$$

Since IRM was designed to estimate global functions, we only use it for this purpose. We introduce two new variants of IRM to estimate environment-specific functions, which we describe next.

- **IRM-PA**: a new variant of IRM for environment-specific prediction functions. It applies the invariance penalty to all parameters. It minimizes:

$$\sum_{e \in \mathcal{E}_{\text{tr}}} R^e(f_e) + \lambda || \nabla_{w|w=1.0} R^e(w \cdot f_e) ||^2 \qquad (3)$$

- **IRM-PP**: a new variant of IRM for environment-specific prediction functions. It applies the invariance

penalty only to the global parameters $\phi$. It minimizes:

$$\sum_{e \in \mathcal{E}_{\text{tr}}} R^e(f_e) + \lambda ||\nabla_{w|w=1.0} R^e(w \cdot f_g)||^2 \quad (4)$$

## 4.1 BASIC SYNTHETIC DATA

We evaluate AIRM on the skin cancer prediction problem described in Section 2. We start with data generated according to the SCM in Example 1. We then generate data according to the SCM in Example 2, the concept drift setting.

For both experiments, we include two training environments: $e_1$ with parameters $\{\sigma_{e_1,x_1} = 1, \kappa_{e_1} = 5, \sigma_{e_1,y} = 1\}$, and $e_2$ with parameters $\{\sigma_{e_2,x_1} = 2, \kappa_{e_2} = 2, \sigma_{e_2,y} = 2\}$. We evaluate on a test environment $e_3$ with parameters $\{\sigma_{e_3,x_1} = 2, \kappa_{e_3} = -2, \sigma_{e_3,y} = 2\}$. $\kappa_e$ is positive for $e_1$ and $e_2$ but is negative for $e_3$. This means that the probability that $x_2 = 1$ is positively correlated with $y$ in the training environments, but negatively correlated with $y$ in the test environments. Thus, models that use $x_2$ will generalize poorly to the test environment. When we generate data for the concept drift setting, we use $\{\mu_1 = 1, \mu_2 = 0, \mu_3 = -1\}$ as the means of the environment-specific $c$ distributions. We set $\sigma_{e,c} = 0.1$ for all environments. We use 10,000 examples from $e_1$ and $e_2$ as training data. We use 1,000 examples from $e_3$ as adaptation data to estimate environment-specific parameters, and 10,000 examples as test data.

We set the hyperparameters: $\lambda = 10,000$, $\eta = 0.01$, and $k_1 = 1$. When implementing AIRM, we compute the estimates of $\beta_e$ directly in phase 2 rather than using SGD, so we don't need to set $k_2$. We report results for 5 random seeds.

We consider linear prediction functions of the form $f_g(\boldsymbol{x}; \boldsymbol{\phi}) = \boldsymbol{\phi}^T \boldsymbol{x}$ (global) and $f_e = f_g + \beta_e$ (environment-specific).

**Example 1 (traditional setting):** The optimal model parameters in this setting are $(\phi_1, \phi_2) = (1, 0)$, since $x_1$ is causal and $x_2$ is spurious. Without concept drift, there is no need for environment-specific intercepts $\beta_e$.

We compare AIRM with ERM and IRM. The results are shown in Figure 2. We see that ERM performs substantially worse on the test data than do the other two methods. The top plot shows that ERM obtains the largest error in estimating each of the $\boldsymbol{\phi}$ parameters. The large value it obtains for $\phi_2$ indicates that it has learned to use the spurious feature $x_2$. In contrast, both IRM and AIRM learn to limit reliance on this feature.

Whereas ERM and IRM are used to train a single global model $f_g$, AIRM learns environment-specific intercepts in addition to $f_g$. This provides it with an extra degree of freedom that is not necessary in this setting. Despite this, we see in the right plot that AIRM obtains performance similar to that of IRM.

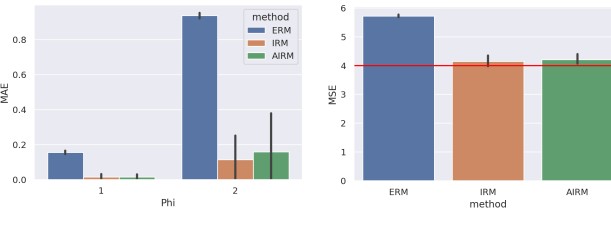

(a) MAE of $\phi$ estimates.  (b) MSE on the test data.

Figure 2: Results of our method (AIRM) and baselines on Example 1 (no concept drift) The red line indicates the MSE of an oracle model $\boldsymbol{\phi} = (1, 0)$.

**Example 2 (concept drift setting):** As in the prior example, the optimal $\boldsymbol{\phi}$ in this setting is $(1, 0)$. With the addition of concept drift, it is now useful to learn environment-specific intercepts $\beta_e$. The optimal value for each $\beta_e$ is $\mu_e$, the mean of the environment-specific context variable distribution.

We present a comparison of AIRM and the baselines in Figure 3. In the right plot, we see that AIRM performs nearly as well on the test data as an oracle that knows the correct model parameters. It outperforms all of the other methods, including IRM and its variants. The left plot shows that AIRM is the only method that is able to learn to avoid relying on the spurious feature $x_2$. The center plot displays the $\boldsymbol{\beta}$ estimation errors. It includes all of the methods except IRM, since IRM does not produce estimates of environment intercepts. We see that AIRM has a low MAE compared to all of the baselines.

While IRM obtains strong performance on Example 1, its subpar performance on this example demonstrates its inability to handle concept drift. In contrast, AIRM performs well in both problem settings.

## 4.2 SYNTHETIC DERMATOLOGY IMAGES

We next consider a slightly more realistic problem. We generate data using the same process as our previous experiments, except that instead of providing $x_1$ and $x_2$ as features, we let $\boldsymbol{x}$ be an image that indirectly encodes $x_1$ and $x_2$. We generate synthetic images of skin lesions using a process inspired by prior work Ghandeharioun et al. [2021]. Examples are shown in Figure 4. We control the size of the lesion based on the lesion abnormality score $x_1$, so that lesions with larger abnormality scores are larger in size. We include markings when $x_2 = 1$ and exclude them otherwise.

We use the same parameters for the training environments and the test environment as the prior example. We use 25,000 examples from $e_1$ and $e_2$ as training data, 1,000 examples from $e_3$ as adaptation data, and 10,000 examples from $e_3$ as test data. We use $\lambda = 100,000$, $\eta = 0.001$, and $k_1 = 1$, and report results for 5 seeds. We again consider linear prediction functions of the form $f_g(\boldsymbol{x}; \boldsymbol{\phi}) = \boldsymbol{\phi}^T \boldsymbol{x}$

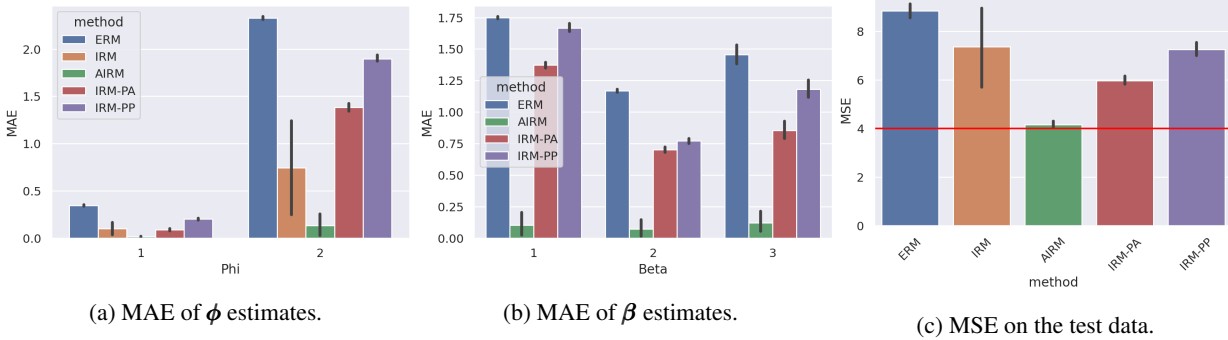

(a) MAE of $\phi$ estimates.

(b) MAE of $\beta$ estimates.

(c) MSE on the test data.

Figure 3: Results of our method (AIRM) and baselines on Example 2 (with concept drift). The red line indicates the MSE of an oracle model $\phi = (1,0), \beta_e = \mu_e$.

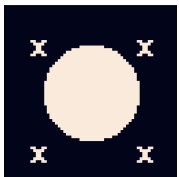 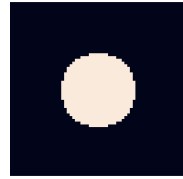

Figure 4: Examples of the synthetic skin lesion images with and without inked markings.

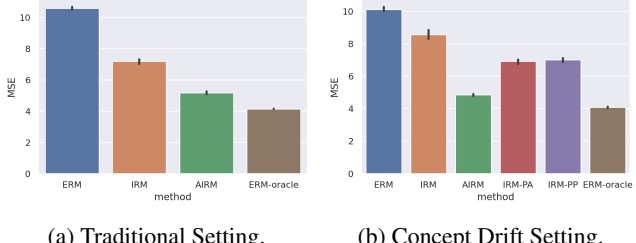

(a) Traditional Setting.

(b) Concept Drift Setting.

Figure 5: Results of our method (AIRM) and baselines on synthetic dermatology image data for the traditional and concept drift settings.

(global) and $f_e = f_g + \beta_e$ (environment-specific).

Since the pixels that encode $x_1$ and $x_2$ vary by image, there isn't a clear optimal setting of $\phi$, as there was in the prior experiment. Instead, as an oracle, we consider a model trained using ERM on the same data, but with the spurious variable (the presence of markings) removed. We present the results on the test data in Figure 5.

**Example 1 (traditional setting):** In the left plot in Figure 5, we see that there is a large gap between the performance of the ERM oracle and standard ERM. This is evidence that the poor generalization performance of ERM results from its reliance on the spurious markings feature. Both IRM and AIRM have better test performance than ERM, indicating that they rely less on the spurious feature. Although these methods show improvements over ERM, their performance is not as close to the oracle as it was in the previous experiment. This suggests that the problem is more difficult when the relevant variables are entangled in a high-dimensional space rather than provided directly.

**Example 2 (concept drift setting):** In the right plot in Figure 3, we again see a large gap between the test performance of ERM and the oracle model, illustrating that a model trained with ERM learns to use the spurious variable. Although the other baselines perform better than ERM, they still have much higher test error compared to AIRM. AIRM is the only method that obtains performance comparable to the oracle model.

# 5 CONCLUSION

Existing invariance-based causal estimation methods assume that the data collection environments do not exhibit concept drift – an assumption that does not hold in many situations. We introduce a new method, called AIRM, that extends IRM to enable invariance-based causal estimation in settings that include concept drift. This work contributes to the broad goal of making machine learning models more robust and causality-aware.

In this study, we examined a limited range of parameter settings. In the future, we plan to conduct a rigorous parameter sensitivity analysis. Since we would like to ground our empirical results in theoretical insight, we plan to analyze the theoretical properties of AIRM, such as its convergence behavior. Finally, we intend to consider a broader set of concept drifts, and to extend the method to address classification problems.

While the results of this initial study are promising, additional work is needed to understand how AIRM performs under a more comprehensive set of conditions. We plan to evaluate AIRM on real-world data, including real-world skin lesion datasets such as Tschandl et al. [2018].

**Acknowledgements**

We gratefully acknowledge the support of Quanta Computer and the Media Lab Consortium Member Companies.

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
