# OpenReview forum: "Invariance-Based Causal Estimation in the Presence of Concept Drift"
_auai.org/UAI/2022/Workshop/CRL — CRL@UAI 2022 Poster_

### Official Review · Reviewer_jLJb · 2022-06-30
**Review for invariance-based causal estimation in the presence of concept drift**

**Rating:** 6
**Confidence:** 5

**Review:**

In recent years, there has been a lot of progress on understanding distribution shifts from the lens of causality. Invariant causal prediction, invariant risk minimization and other works in this branch of literature rely on invariance principle. Invariance principle states that the so long as we do not intervene on the target, the distribution of the target conditional on the causes remains invariant across all the interventions. In most existing works, the authors assume that all causal variables can be inferred from the data. In this work, the authors consider the setting when some of the causes may not be obsered. The authors propose an alternating IRM algorithm, which alternates between IRM and an intercept adjustment phase.  I think the Assumption 2b feels quite strong and the authors should consider relaxing it in an extension of the work. The authors have ignored some works that also try to move beyond the standard application of invariance principle, see https://arxiv.org/pdf/2107.01876.pdf and http://proceedings.mlr.press/v130/ahuja21a/ahuja21a.pdf.
Overall, I found the work clearly written, it has reasonable degree of originality, and can be of interest to researchers in the area.

---

### Official Review · Reviewer_fSNy · 2022-07-05
**Fixing concept drift for IRM**

**Rating:** 5
**Confidence:** 4

**Review:**

This paper proposes a new method called Alternating Invariant Risk Minimization to account for potential concept drift between different environments. The motivation for this approach is the fact that in many real-world scenarios the label distribution is environment dependent. Given this motivation, it is disappointing that the authors only provide experimental validation of their method on synthetic toy datasets which makes it very difficult to conclude whether this method will be useful in any real-world setting.
Notwithstanding that, the paper is easy to follow and the method is described in enough detail making it easy to understand.

---

### Meta-Review · Program_Chairs · 2022-07-06

**Recommendation:** Accept (Poster)
**Confidence:** 2

**Metareview:**

Reviewers raised concerns on absence of real-application, lack of discussion of some related work and on certain assumptions of the method. Nevertheless, the contribution appears clearly stated and may have the potential to spark interesting discussions. The authors are encouraged to integrate the reviewer's suggestions as much as possible for a future version. For future versions of their work, they are additionally encouraged to consider real-world applications beyond the toy settings discussed in the current version.

---

### Decision · Program_Chairs · 2022-07-06

Accept (Poster)